# Understanding the implementation, impact and sustainable use of an electronic pharmacy referral service at hospital discharge: A qualitative evaluation from a sociotechnical perspective.

**Mark Jeffries**[1,2]*, **Richard N. Keers**[1,2,3], **Hilary Belither**[4], **Caroline Sanders**[2,5], **Kay Gallacher**[6], **Fatema Alqenae**[1], **Darren M. Ashcroft**[1,2]

**1** Centre for Pharmacoepidemiology and Drug Safety, Division of Pharmacy and Optometry, School of Health Sciences, University of Manchester, Manchester, United Kingdom, **2** NIHR Greater Manchester Patient Safety Translational Research Centre, University of Manchester, Manchester Academic Health Sciences Centre (MAHSC), Manchester, United Kingdom, **3** Pharmacy Department, Greater Manchester Mental Health NHS Foundation Trust, Manchester, United Kingdom, **4** Salford Royal NHS Foundation Trust, Salford, United Kingdom, **5** Division of Population Health, Health Services Research & Primary Care University of Manchester, Manchester, United Kingdom, **6** Patient and Public Involvement, NIHR Greater Manchester Patient Safety Translational Research Centre, University of Manchester, Manchester Academic Health Sciences Centre (MAHSC), Manchester, United Kingdom

* mark.jeffries@manchester.ac.uk

**Data Availability Statement:** We have provided supplementary files of the coding frameworks. In

## Abstract

### Introduction

The transition of patients across care settings is associated with a high risk of errors and preventable medication-related harm. Ensuring effective communication of information between health professionals is considered important for improving patient safety. A National Health Service(NHS) organisation in the North West of England introduced an electronic transfer of care around medicines (TCAM) system which enabled hospital pharmacists to send information about patient's medications to their nominated community pharmacy. We aimed to understand the adoption, and the implications for sustainable use in practice of the TCAM service

### Methods

We evaluated the TCAM service in a Clinical Commissioning Group (CCG) and NHS Foundation Trust in Salford, United Kingdom (UK). Participants were opportunistically recruited to take part in qualitative interviews through stakeholder networks and during hospital admission, and included hospital pharmacists, hospital pharmacy technicians, community pharmacists, general practice-based pharmacists, patients and their carers. A thematic analysis, that was iterative and concurrent with data collection, was undertaken using a template approach. The interpretation of the data was informed by broad sociotechnical theory.

addition, we have provided a supplementary file of all the extracts from the transcripts detailed per theme and code from the final coding. This extensive and detailed document provides a data set from which the study can be fully replicated. Full transcripts are not available to preserve the anonymity of participants as per our ethics. This is a qualitative study confined to relatively small groups of health care professionals in specific roles and patients. Making the full data set publicly available could therefore potentially lead to the identification of participants. Our ethics approval was granted based on the anonymity of the individuals consenting to participate and specifically referred to only anonymised quotations from transcripts being made available as we have in the supplementary file. As such the participants did not consent to full their transcript being made publicly available. Ethics approval was granted by North West –Greater Manchester East NHS Research Ethics Committee, 3rd Floor, Barlow House, 4 Minshull Street, Manchester, M1 3DZ.

**Funding:** The study is funded by the National Institute for Health Research through the Greater Manchester Patient Safety Translational Research Centre (NIHR Greater Manchester PSTRC) grant number PSTRC-2016-003. The views expressed are those of the author(s) and not necessarily those of the NHS, the NIHR or the Department of Health and Social Care. The funders had no role in study design, data collection and analysis, decision to publish, or preparation of the manuscript.

**Competing interests:** The authors have declared that no competing interests exist.

## Results

Twenty-three interviews were conducted with health care professionals patients and carers. The ways in which the newly implemented TCAM intervention was adopted and used in practice and the perceptions of it from different stakeholders were conceptualised into four main thematic areas: The nature of the network and how it contributed to implementation, use and sustainability; The material properties of the system; How work practices for medicines safety were adapted and evolved; and The enhancement of medication safety activities. The TCAM intervention was perceived as effective in providing community pharmacists with timely, more accurate and enhanced information upon discharge. This allowed for pharmacists to enhance clinical services designed to ensure that accurate medication reconciliation was completed, and the correct medication was dispensed for the patient.

## Conclusions

By providing pharmacy teams with accurate and enhanced information the TCAM intervention supported healthcare professionals to establish and/or strengthen interprofessional networks in order to provide clinical services designed to ensure that accurate medication reconciliation and dispensing activities were completed. However, the intervention was implemented into a complex and at times fragmented network, and we recommend opportunities be explored to fully integrate this network to involve patients/carers, general practice pharmacists and two-way communication between primary and secondary care to further enhance the reach and impact of the TCAM service.

## Introduction

Medication safety during transition from hospital to home is a key priority of the World Health Organization's Third Global Patient Safety Challenge (2017) "Medication without harm" [1]. A recent systematic review reported that across 54 studies a median of nearly half of adult or elderly patients had a medication error or unintentional medication discrepancy following hospital discharge, with a median of one in five of these patients affected by adverse drug events [2]. Older patients may be at heightened risk of patient safety incidents [3, 4] and particularly medication related harm following hospital discharge due to the likelihood of frailty, multimorbidity and associated complex medication regimens [5]. In a United Kingdom (UK) study by Parekh et. al, of 1280 older adults followed up eight weeks post discharge from hospital, 413 (37%) of participants experienced medication related harm of whom 323 (78%) experienced a serious event with 4 (1%) dying as a consequence [6].

The effective communication of information between different health professionals across primary and secondary healthcare boundaries is considered important for improving patient safety. The ongoing use of medicines by patients and their carers might also be impacted by the quality of this process where failures in communication can lead to non-adherence or medication errors which as a consequence can lead to medication related harm incurring hospital readmission [1, 7]. Changes in patient's medication when in hospital can result in discrepancies between the list of medicines held in primary care and those provided to the patient on discharge [8, 9]. Patients may also lack knowledge and information about changes in medicines at the time of hospital discharge [10] and attribute this to poor explanations from health

professionals, a lack of information provision and lack of patient involvement in relevant decision making [11]. The quality of communication between secondary and primary care may be variable, with hospital discharge summaries often lacking important information such as changes in medications, co-morbidities, allergy status or diagnostic test results [7, 12–14]. It has also been suggested that the communication of discharge summaries to community pharmacy has been inconsistent, incomplete and lacked timeliness and that the potential for community pharmacy involvement in the discharge process is underutilised [15].

Efforts to improve medication safety at care transfer have included pharmacist-led interventions at both hospital and in the community, with information technology being utilised to improve accuracy of discharge summaries including to community pharmacy [7, 14, 15]. Enhanced pharmacist-led transitions of care services in both hospital and community settings have been reported to reduce readmissions, and lower rates of ADEs and health costs [16–18]. Within hospital settings interventions by pharmacists such as patient education at discharge, discharge planning and post discharge follow–up has been found to increase patient knowledge about their medicines [19], decrease hospital readmission and support the resolution of medication discrepancies to reduce preventable adverse drug events [20, 21]. Community pharmacists receiving discharge information has been seen to effectively reduce discrepancies between the General Practitioner (GP) medication record and the hospital list and between patients' self -described medication regimen and their hospital discharge letter [21].

The NHS Discharge Medicines service has recently been introduced into community pharmacy as an essential service as part of the Community Pharmacy Contractual Framework in England [22]. This provides a toolkit for community pharmacy and NHS hospital trusts to ensure that there are integrated approaches to medicines optimisation for patients who have been discharged from hospital. In England Academic Health Science Networks (AHSNs) are regional networks of health, academic, local government partners who can lead on delivering innovations at scale. The AHSNs have been instrumental in the implementation and roll out throughout England of Transfer of Care Around Medicines (TCAM) projects which allow for electronic messages to be sent from hospital pharmacists to a specific community pharmacy on patient discharge [23–25]. TCAM services are considered a medicines safety intervention for patients who have been identified as needing support for their medicine taking or who may be at risk of adverse events following discharge from hospital. [22] Wilcock et al [26] explored through a cohort study, the readmission rates from transfer of care referrals to community pharmacy after discharge and found a lower rate of readmission in those who had an actioned transfer of care service from the community pharmacy. Previous research on TCAM services indicated how electronic transfer of discharge information to community pharmacists has aimed at improving the continuity of care for patients [27]. Patients who had such a pharmacist follow up were less likely to be readmitted to hospital, but very few patient referrals were accepted or followed-up in the community pharmacies [27]. In a similar electronic referral service, patients discharged were directly referred to community pharmacists for a post discharge medicines consultation or referral including Medicines Use Reviews and New Medicines Service into other appropriate care pathways [28]. A qualitative evaluation of this service in the North-West of England from the perspectives of health professionals found that whilst professionals sharing views of the benefits of the intervention helped with implementation, there were barriers such as information and training which limited implementation in community settings [29].

There is an increasing utilisation of information technology for medication safety. However, the implementation of such technology has had varied success [30, 31]. The implementation of information technology in health care settings has been explored from socio-technical perspectives which explore not just the ways technology is used but the social and organisational contexts into which it embedded [32, 33]. Sociotechnical theory considers that people

and technology are reciprocally and recursively related and as a consequence the outcomes of the relationships between the social, human agency and technology are considered as interdependent and not simply as the interactions between homogenous unique elements [34]. These sociotechnical approaches emphasize that contextual factors, technology, and human agents are dynamically connected and operate in multiple ways [32] and focus upon the social processes involved in the use of the technology. Technology has been understood as offering the possibility for different actions and has therefore been conceptualised as offering affordances [35] Such affordances may operate within a combination of the material properties of the technology and social processes, a concept understood as sociomateriality. In sociomateriality the technology may have fixed properties which allow for differences in use to be ascribed to the context and social processes within which the technology is used [36]

Further qualitative evaluation is therefore needed to uncover the ways in which the new TCAM services are implemented and used, how networks of social relations might impact upon the delivery of the service, including networks that involve patients and carers, and upon how communication of information through the service might enhance medication safety activities. This current qualitative evaluation focused upon one particular TCAM intervention enabling hospital pharmacists in an NHS hospital trust to electronically transfer discharge information to local community pharmacists via the PharmOutcomes™ platform for post discharge medicines optimisation. This service focused specifically upon those patients in receipt of a new or existing monitored dosage system (MDS). Monitored dosage systems are in the form of prepacked boxes or blister packs with medicines organised by day and time. These are commonly used to dispense medicines for patients who may be prescribed multiple medicines and who may experience difficulty in maintaining a routine of medicine taking. The TCAM service gave the opportunity for additional community pharmacy support of patients informally or through paid NHS services such as the Medicine Use Reviews (MUR) or New Medicines Service (NMS). Additionally, a Neighbourhood Integrated Practice Pharmacist (NIPPS) service had been implemented by the hospital trust in 2016 in which clinical pharmacists were employed to work in general practices to optimise medicines use. The relevant NIPPs service lead pharmacist had oversight of referrals in their area, allowing them to facilitate communication between the GP surgery and community pharmacists for timely resolution of issues. In this study it was assumed that the implementation, use and sustainability of the electronic TCAM system would be dependent on a range of factors including the complex network of social interactions between actors, organisational norms, institutional work practices, and through existing and new infrastructure. We aimed to understand from the perspectives of health professionals, patients and carers the implementation, impact and sustainable use of this electronic TCAM system.

The electronic TCAM service was evaluated in Salford (Greater Manchester, UK) Clinical Commissioning Group (CCG), which has a population of 270,000, and Salford Royal NHS Foundation Trust, using a qualitative design. We undertook semi-structured interviews with a range of health professional stakeholders in order to understand how the contextual background, including work practices and socio-organisational processes influenced the various ways in which the TCAM service was implemented, executed in everyday use, and sustained. In interviews with patients and carers we explored understanding of the service perceptions of the potential impact upon patients and medication safety, and broader medicines use.

## The intervention

The Transfer of Care Around Medicines at Salford Royal Foundation Trust (SRFT) service was specifically designed to provide community pharmacists with hospital admission and

discharge information about their patients who were in receipt of a Monitored Dosage System (MDS).

**On admission** the hospital pharmacist, during drug history, identifies if the patient is in receipt of an MDS. If so an order for 'Community Pharmacy Referral' is activated within the hospital Electronic Patient Record (EPR). This then initiates an electronic message to be sent to the patient's nominated community pharmacist through PharmOutcomes to notify them that the patient was in hospital. Within the community pharmacy the referral is accepted or rejected at this stage. If the patient is known to the community pharmacy they should accept the referral at discharge they can then check the discharge summary for changes in medications.

**Upon discharge** the pharmacist checks same 'community pharmacy referral order' in the EPR and adds any messages as appropriate (eg. if this is a new MDS). Two hours after the patient is discharged a referral message indicating the discharge is automatically received via the PharmOutcomes platform at the patient's nominated community pharmacy. The full discharge summary is attached to this referral message. If subsequent changes are made to the discharge summary up to 10 days post discharge a new document is automatically attached to the referral message to alert the community pharmacist to these.

## Sampling and recruitment

Participants were opportunistically sampled from those health professionals working in secondary and primary care (community pharmacists, hospital pharmacists, hospital pharmacy technicians, general practice-based pharmacists and neighbourhood leads) who were actively involved in the implementation or use of the pharmacy e-referral TCAM system. The aim here was to find a range of different professionals to explore a range of views and perspectives of the service. Additionally, patients and carers for whom referrals had been made during their hospital admission were opportunistically sampled. The health professionals were approached through a range of professional pharmacy networks known to the researchers through the National Institute for Health research (NIHR) Greater Manchester Patient Safety Translational Research Centre (GM PSTRC) Community Pharmacy Patient Safety Collaborative and Salford Royal NHS Foundation Trust (SRFT). Participants returned consent-to-contact forms if they wished to take part. They were then approached by telephone, email, letter or face-to-face by a member of the research team (HB or MJ). Three health professionals declined to take part or did not return consent to contact forms. Those who declined cited workload and time commitments as the reason for their non-participation.

Patients or carers were identified through hospital pharmacists involved in the patients' care and approached whilst in the hospital. Potential patient participants were provided with written information about the study alongside being provided information about their TCAM referral. Patients were not approached if they were confused, unconscious particularly unwell or frail and therefore in the opinion of the pharmacist unable to give informed consent. If during the approach by the hospital pharmacist, the patient indicated they would prefer their carer to take part they would then be asked to forward the information to their carer. Carers who agreed to take part returned consent-to-contact forms to the research team. All potential participants were given at least 24 hours to consider study information. Each participant in the study was assigned an identification number. This number, rather than the participant's personal details, was used to identify any interview data associated with the participant.

## Data collection

The semi-structured interviews with health professionals took place in private at the participant's usual place of work (17) or on university premises (1). Interview topic guides were developed by the research team and were designed to elicit views and perceptions of using the

electronic TCAM system (see S1 Appendix). This included the benefits and drawbacks of the service; how organisational norms, work practices, workflow and conventions interacted and impacted upon the ongoing use of the service; and the communication and collaborations between health professionals. This would help us to understand the factors that might influence the acceptability, implementation, sustainable use and potential impact of the service. The interviews with patients and carers took place in private at the participant's home and were designed to elicit perceptions and views following experience of the service. This included patient and carer perceptions of how they were given information about their medicines from health professionals and how they understood any interactions with the pharmacist. Additional questions focused on medicine related problems the including the supply and availability of their medicines, the perceived feasibility of the service, the appropriateness of referrals and how the service potentially supported patient and carer understanding of their medications. All carers were family members who informally cared for the patient. Interviews with health professionals lasted between 29 and 52 minutes. The interviews with patients and carers lasted between 22 and 42 minutes. Interviews were conducted between May 2019 and December 2019 by MJ, an experienced qualitative researcher (see Table 1). Health professionals

**Table 1. Interviews and participants.**

| Interview | Interview length (mins) | Interview–Individual, dyadic or triadic | Participant (s) | Participant role |
|---|---|---|---|---|
| 1 | 42 | Individual | HP1 | GP Practice Based Pharmacist |
| 2 | 36 | Individual | HP2 | Hospital Pharmacy Tech |
| 3 | 36 | Individual | HP3 | Hospital Pharmacist |
| 4 | 42 | Individual | HP4 | Hospital Pharmacist |
| 5 | 36 | Individual | HP5 | Hospital Pharmacy Tech |
| 6 | 29 | Individual | HP6 | Hospital Pharmacy Assistant |
| 7 | 31 | Individual | HP7 | Hospital Pharmacist |
| 8 | 36 | Dyadic | P1 | Patient |
| | | | C1 | Carer |
| 9 | 40 | Individual | HP8 | GP Practice Based Pharmacist |
| 10 | 41 | Individual | HP9 | GP Practice Based Pharmacist |
| 11 | 42 | Individual | HP10 | Community Pharmacist |
| 12 | 32 | Individual | HP11 | Community Pharmacist |
| 13 | 41 | Individual | HP12 | Community Pharmacist |
| 14 | 32 | Individual | HP13 | Community Pharmacist |
| 15 | 52 | Individual | HP14 | Hospital Pharmacist |
| 16 | 34 | Individual | HP15 | Community Pharmacist |
| 17 | 48 | Individual | HP16 | Community Pharmacist |
| 18 | 30 | Individual | P2 | Patient |
| 19 | 43 | Individual | P3 | Patient |
| 20 | 22 | Dyadic | C2 | Carer |
| | | | C3 | Carer |
| 21 | 42 | Triadic | C4 | Carer |
| | | | P4 | Patient |
| | | | C5 | Carer |
| 22 | 41 | Individual | HP17 | Community Pharmacist |
| 23 | 39 | Individual | HP18 | Community Pharmacist |
| Total 23 | Average 37mins Range 22mins to 52 mins | | | **GP Practice Based Pharmacists = 3; Community Pharmacists = 8; Hospital Pharmacist = 4; Hospital Pharmacy Technician = 2, Hospital pharmacy assistant = 1; Patients = 4; Carers = 5** |

received £50 per hour in shopping vouchers to reimburse their time. Patients and carers received £20 per hour in shopping vouchers. Interviews were digitally audio-recorded and fully transcribed verbatim.

## Ethics approval and consent to participate

Ethical approval for the study was granted by the North-West–Greater Manchester East NHS Research Ethics Committee (19/NW/0110). All interview participants gave written informed consent to take part in the study, and for the interviews to be digitally audio recorded and transcribed verbatim. Health professionals and carers who agreed to take part completed written consent at the start of the interview. Written consent was taken from patients by the hospital pharmacist, and they were then contacted by a member of the research team to arrange the interview.

## Data analysis

Following transcription, anonymised interviews were organised using QSR NVIVO® Pro v12 software. We undertook a thematic analysis informed by Braun and Clark [37]. Analysis followed an iterative approach and was concurrent with data collection. This allowed for the development of the coding framework and for emergent findings to be explored in subsequent data collection. Initial inductive coding was followed by a template approach through the development and refinement of coding frameworks [38]. MJ read each transcript in a process of immersion. A selection of early interviews was read and discussed by MJ, RNK, KG, HB and FA. MJ inductively coded a sample of six transcripts focusing upon the interactions of different people in the network, the social processes within the intervention and the changes brought about by the technology. Identifying these features, and patterns allowed for groups of codes and potential themes to be refined into a coding template with codes grouped into sets (see S2 Appendix). This coding template and further coded transcripts where then discussed by MJ, RNK, KG, HB and FA. From these discussions the template was revised and refined into themes and codes (see S3 Appendix). This template was then applied to the full dataset. It was from this final stage of the data analysis that the final themes and sub-themes were interpreted (see Table 2).

## Results

Twenty-three interviews were conducted with health care professionals (n = 18 participants, 18 interviews) and patients and carers (n = 9 participants, 5 interviews). Two interviews with patients were conducted one-to-one with the remaining three as group interviews (patient/carer; carer/carer; patient/carer/carer). See Table 1

**Table 2. Themes and sub themes.**

| Main Theme | The nature of the network and how it contributed to implementation, use and sustainability. | The material properties of the system. | How work practices were adapted and evolved | The enhancement of medication safety activities |
|---|---|---|---|---|
| Sub Themes | Relationships between different health professionals and what this achieves | The previous system and changes in technology | Adaptation of work processes to the availability of information | Enhanced clinical time |
| | Communication and movement of information in the network | The new technology provided new infrastructure | Timesaving, speed, and efficiency | The avoidance of mistakes in transition and improved accuracy of medicine reconciliation |
| | Patients and carers | | | |

The ways in which the new e-referral system was adopted and used and the perceptions of it from different stakeholders were conceptualised into four main thematic areas (see Table 2):

- The nature of the network and how it contributed to implementation, use and sustainability.

- The material properties of the system.

- How work practices were adapted and evolved.

- The enhancement of medication safety activities.

## The nature of the network and how it contributed to implementation, use and sustainability

The new TCAM service required an interdependent, collaborative network of different stakeholders in order for the communication of discharge information to impact upon care with medicines. The service, through the increased sharing of information, led to a building of relationships between different health professionals in secondary and primary care, including between general practices and community pharmacists. These relationships supported the standardisation and streamlining of information exchange between hospital and community pharmacy. Patients or their cares actively managed their medicines and initiated informal communication with community pharmacists and GPs. However, where the network was incomplete, and fragmented the effectiveness of the intervention was diminished and therefore the movement and sharing of the information across the network was hampered. There was no two-way communication back to the hospital pharmacists, communication between general practices (either with GPs or practice-based pharmacists) was uneven and incomplete across the network and communication between health professionals and patients relied upon the active role of patients and their carers.

**Relationships between different health professionals and what this achieves.** A number of different stakeholders were involved in the network. The availability, accessibility and quality of the information impacted upon sharing across this network. Where participants reported communication was good, they felt this could improve and develop relationships across the network which then could lead to further improvement in communication. This was particularly seen in communication between community pharmacy and general practice, particularly through practice-based pharmacists. However, the system did not provide functionality for information exchange back and forward between community pharmacy, general practice and hospital. As one community pharmacist reflected such communication in the future could provide a 'joined-up' approach in sharing how they hadeach talked to the patient.

*One joined up approach so that if that patient then went back in the hospital pharmacist could see, oh, there've been some conversations here, so I can see the community pharmacist talked about your inhalers, [. . .]it's a better conversation for the patient, for the pharmacist, it's more valuable (Community pharmacist_HP16)*

**Communication and movement of information in the network.** The TCAM allowed for the movement of discharge information across the network. This was particularly valued by this hospital pharmacy technician in that he felt assured the information was being received in the community pharmacy.

*So, the beauty of the e-referral, [. . .], is that firstly we get to inform the community pharmacy that the patient is admitted into hospital, they get to know that electronically. And the second*

*thing is [. . .] once an e-referral has been setup they get a discharge summary once the patient goes. So, it takes the pressure off of me, trying to remember whether or not we've informed community pharmacists [. . .]* **(Hospital pharmacy technician_HP5)**

The enhanced information available to community pharmacy was perceived as allowing for an *'extra check'* **(Community pharmacist_HP12)** of the discharge summary and any changes made in patient's medicines during the transfer of the information through the network. Participants attached importance to involvement from general practice and the patient in this checking process.

*I know there has been a couple of occasions previously where we've had discharge prescriptions from the surgery, we haven't actually seen the discharge summary, we've just been told, right this patient is out of hospital this is the new scripts, done them, sent them out to the patient and the patient says, 'I've not got everything here, I was on a lot more medication' and [. . .] half of the stuff hasn't been done for some reason. Whereas now there is that extra check because we are seeing the discharge summary, the doctor is seeing the discharge summary, the patient and family, depending on how able they are to deal with it.* **(Community pharmacist_HP12)**

There was recognition that the network was important to moving information around to ensure that all health professionals caring for the patient had the correct information and to ensure, as this hospital pharmacist described, the community pharmacy were made aware that the patient was in hospital.

*. . .. it's like a chain, isn't it, I guess, it's like if one part is not working and the rest isn't, so if you don't put the referral in the first place community pharmacy is not going to know they're in hospital. But if they don't pick it up, if they're not proactive in picking up the referrals, then it's not going to work either.* **(Hospital pharmacist_HP4)**

The network could become fragmented if the information was ineffectively moved through it or if the relationships between different parts were broken or undermined. This community pharmacist reflected upon the different approaches to prescribing of two different GP surgeries with one surgery having pharmacists, and a prescription clerk and a detailed division of labour and the other without any clear leadership around prescribing.

*They will have a pharmacist or two pharmacists in charge of running it, they will have prescription clerks, all these people are doing all activities with regards to prescribing prescriptions, discharges, you name it[. . .] The next door doctors, unfortunately, it was all over the place, in my view. There was usually a doctor leading this, signing, but there was no one really taking full charge of every single person was issuing. It was utter chaos.* **(Community pharmacist_HP10)**

**Patients and carers.**   Patients appeared to be unaware of the network and the technology. Patients reported lacking knowledge of the TCAM intervention system and of their medicines, and some felt explanations of these were needed but had not been given. One patient did not know why they took their medicines or what they were for;

*"I just take them, you know, and I don't know what they're for [. . .] I mean, I take all these, and I haven't got a clue what they're for"* **(Patient_P2)**

This suggested that the TCAM service was not something patients were aware of despite having been referred, as all patients who were interviewed were. The service may not have been supporting increased knowledge of medicines amongst patients. This was attributed to a lack of connection between health professionals and patients, and as one carer suggested the need to initiate communication.

*"But they don't speak to us about it [medications]. It's only because I got in contact and said can we change any particular meds. So, it was only my instigating it."* **Carer _C1**

Patients suggested that they preferred face-to-face communication with health professionals but often this took place by telephone typically through patients or carers calling the GP surgery. Similarly, one patient talked of receiving explanations only after asking questions and added *'if I don't ask, I don't know'* **(Patient _P3)**

Patients and carers also felt the need to inform the pharmacy that they or their relative had been discharged from hospital and be proactive in ensuring that a new supply of medicines was dispensed. As one patient reflected there had been an occasion when the pharmacy had not had the discharge information or been able to supply the new medicines on time. This meant that the patient had their old MDS and additional new medication in boxes which presented a safety risk.

*When I left hospital, the day after I went in to the pharmacy[…], they said, we've had a phone call from the hospital pharmacy and they have told us that there's some changes, so I said, so you've not had the discharge papers, so they said no, I said, have you had the new list of medication, no, have you heard from the surgery, no. […] they said that we haven't had time to get you one ready, your next one [supply of medication], she said, so the one [supply] that we've got ready has got all your old meds on and there's no changes, so we'll have to give you that and the boxes of the additional tablets….(Patient _P3).*

A limitation in the communication with patients was highlighted by a community pharmacist who explained that the patients who were in receipt of a MDS, for whom the TCAM intervention was designed, were less likely to visit the pharmacy to collect their medicines, so opportunities for further pharmacy support were lost.

*a vast majority of our patients that are on (MDS) we deliver to […] they don't come, so we can't–we do have quite a few that collect, so we do the new medicine service with them and give them something relevant.[…] But I think it does, not necessarily give you an official MUR or an NMS but it gives you that opportunity for a discussion with the patient, are they happy with how things are, do they understand what changes they've got…(Community pharmacist_HP12)*

As a consequence of the TCAM intervention participants reported that there was increased engagement with patients because the pharmacist had more information about their hospitalisation and changes in their medicines. On delivering the medicines to the patient at their home they were able to discuss the changes with them. Patients were said to find these conversations helpful.

*And then also when I go to the patient with it […] if there are changes I do a medicine review with them all about the changes to let them know […] in their home, yeah, what it is about and I would not have done that before. […] I've found that that's helpful, very helpful and I*

*think the patients find it helpful, they've all been very grateful because maybe they haven't understood what the doctor had said to them, maybe they didn't know that their medication was changing or they weren't listening. No, I really didn't do that much before.(Community pharmacist_HP12)*

## The material properties of the system

**The previous system and changes in technology.**   The previous discharge system was reported to involve variable communication and work practices that were dependent upon the individual, and the sharing of discharge information was often informally by telephone or fax machine. Changes in work practices took place directly as a consequence of the new e-referral system and meant staff no longer needed to use fax machines and or to telephone surgeries or the hospital for information,

> *I: What was there before this, in terms of any communication?*
>
> *R: Nothing. A telephone, yeah. So, I know, 'cause I've worked in community, and obviously I'm working in practice now. I've never worked in hospital. But I know that, when I was in community, we would get the discharge faxed from the practice. (Practice-based pharmacist–- HP1)*

The changes in technology that the system brought introduced a new infrastructure and new material properties to support and enable the work of pharmacists. Hospital pharmacists found the new system streamlined and provided a stepwise approach to sending the electronic referrals which involved documenting patient medication history and prompting an TCAM referral if required. This was seen as preferably because it provided greater accuracy, ensured that those patients in receipt of an MDS were referred. The automatic nature of this was perceived as ensuring that the pharmacist did not forget to complete the referral.

> *It's an electronic document which pharmacists and technicians use [. . .] it's all about accurately documenting the drug history for a patient, [. . .] And then you document the drug history, so you write whatever they're on at home, and then it says [. . .], but it says something like, can you complete an e-referral, do they have a compliance aid [MDS] basically, and if you tick yes, it automatically brings up the box for the e-referral, so you can click on that and it exports you to [. . .] takes you straight through to the e-referral, which is good because it means you're less likely to forget to do it. (Hospital Pharmacist_HP4)*

Such information technology infrastructure was seen as more efficient, more effective, and safer than previous systems that had involved faxing. The fax system required a person to process them, was perceived as unreliable and did not always supply the full information through technical malfunction.

> *. . .used to be by fax until this new system came, and that used to be a nightmare [. . .] . . .fax is not really reliable and doesn't always work and the number's busy and sometimes they don't receive the full fax. So that causes a lot of issues and they always used to complain to the hospital that they'd not received it but then it's not always the hospital because they might have done their part but it's just not got through,. (Practice-based pharmacist_HP9)*

**The new technology provided new infrastructure.**   The TCAM service provided an infrastructure that enabled the availability and accessibility to community pharmacists of more

enhanced information regarding both the admission of patients to hospital and their discharge back to the community. Participants commented that previous exchange of information had been unsystematic, fragmented and utilised outmoded technology. In the former system, community pharmacists were not routinely informed that patients had been admitted to hospital.

*The fact that we actually get told when one of our patients goes into hospital, is a big thing. Because quite often they'd ring and say I'm out of hospital, they've changed my medicines, we didn't even know you were in, we've still been sending you medication every week, where's that been going, you know. And to get the discharge information as well. 'Cause sometimes we would get that faxed through, sometimes we wouldn't. They went through a spell where they wouldn't fax it, trying to get it off the doctors sometimes is difficult.* **(Community pharmacist_HP12)**

The information was perceived, particularly in community pharmacy, as of enhanced quality than previous systems in that it was more accurate, arrived in a more timely manner and provided *"a clearer discharge letter"* **(Community pharmacist_HP15)**. Community pharmacists valued the clarity and completeness of the information, knowing one of their patients was in hospital and the greater accuracy and reliability of the discharge information they received. This was particularly the case now that the discharge information was electronic as opposed to handwritten letters that were faxed. The handwritten notes were perceived as unclear, and this could lead to confusion and inaccuracy in changes made to the medicines.

*So, it gives a clearer discharge letter to us first of all. So, the handwritten ones, sometimes they are not very clear and it causes confusion. But when it's electronic, it's easier for us to go through and compare to the old medication and see if there are any changes. So, it's quicker for those purposes. Also, because it is electronic, we get to know quicker than we would usually if the patient has been admitted to the hospital or is discharged.* **(Community pharmacist_HP15)**

Community pharmacists valued receiving the same information as general practices following hospital care. The extra information provided by the TCAM system, including the reason (s) why the patient had been admitted to hospital which helped community pharmacists to form a more complete picture of the patient's status, diagnosis and medicine changes where previously faxed information had not provided the full picture.

*So, you get the full picture. So, you'll read the discharge sheet [. . .] and you get the full picture of the patient, you get to look into it more, you get more detail. Just a list of meds is okay, so you can match the meds against the scripts or chase it up or look for what's going on, but when you've got the full picture of any contraindications, medical history, any medicines that were stopped and started, any that were in hospital but have then been stopped, you've got all the extra detail on [. . .] there's no guesswork. You know what's gone on when they've been in hospital, you know what's been stopped. . .. **(Community pharmacist_HP17)***

## How work practices were adapted and evolved

In order to facilitate the transfer of information through the network and support subsequent action to safeguard patient care, work practices were adapted.

**Adaptation of work processes to the availability of information.**   The new information and associated technology changed work practices, some of which were related to time saving

but others were related to changes in the ways people worked with different technologies. The introduction of the technology 'w*as fairly self-explanatory' (Community pharmacist_HP12)* and one hospital pharmacist talked of being well prepared for changes through training

> *We had training by two of the pharmacists within the hospital [. . .] and they went through the whole system with us, explained what, you know, was going to happen, what were we to do if we had any problems, to contact one of them too. [. . .] when it went live, we were all pre-pared, we all knew what to expect.* **Hospital pharmacy technician _HP2**

The admission notification that was sent to the community pharmacy led to a specific and clear new workflow as described by this pharmacist. This meant that for community pharmacy there were new processes involved in ensuring they did not make up the MDS that they did not undertake previously.

> *an email comes through [. . .] and that flags as, there's a message on PharmOutcomes. So, then we'll log into PharmOutcomes and it'll say, we've got an inpatient. So, one of our patients is. . .that's on a tray is now in hospital. [. . .]Put them on the hospital shelf. Put the printout from PharmOutcomes in that box. Put everything on hold. (Community pharmacist_HP17)*

Changes and adaptations to the technology, facilitated by the TCAM system, provided further infrastructure which led to changes in work practices. Community pharmacies were issued with an alert system that they could plug into their computer and be given real time alerts of referrals without having to check emails.

> *We have now got the Pharmalarm as well [. . .] so it's like a little plug in alarm thing that basi-cally changes colour when we get any messages or any notifications. So, you still get the email as well. So, sometimes it will change and start flashing blue when we haven't even noticed and we just see the email first. It's just like a little widget that changes colour and flashes [. . .] so it lights up lighter initially if we've got any discharge referrals it will flash blue (Community pharmacist_HP12)*

**Timesaving, speed, and efficiency.** The TCAM intervention was reported to be a more streamlined system that allowed community pharmacists more time to process prescriptions. This was seen to reduce rushing which was seen as a safety issue. Pharmacists talked of having *'a lot more time to sort everything out' (Community pharmacist_HP12).* The timely arrival of the discharge was seen as being much better for the patient.

> *At that point the patient's got a week's worth of tablets, right, therefore you do it quickly and reconcile it when they need a prescription, you've at least got a week [. . .] To get it, you know, and we're getting. . .that's, you see, very helpful [. . .]. This is much better for the patient, much better, definitely.*

## The enhancement of medication safety activities

**Enhanced clinical time.** For both hospital and community pharmacists and technicians, time spent on administrative duties was reported to be reduced and clinical time perceived to increase as a consequence of the TCAM service. For community pharmacists completing clini-cal checks on medications and engaging with patients was facilitated and enhanced by the availability of the information accessed through the TCAM intervention allowing community pharmacists to complete informal or formal reviews patients' medicines:

*So that means again our conversations are more technical, they're more about the clinical, the issues rather than actually trying to do, in essence, admin work, trying to just get the information so that we can try and do something with it. (Community pharmacist_HP10)*

Similarly, for this hospital pharmacist the time saved allowed her to engage more with patients face to face.

*It allows me time to actually review the ward and look at other patients who are probably more sick, other patients that have been in longer. It just allows me more time to do other ward work. (Hospital Pharmacist_HP3).*

**The avoidance of mistakes in transition and improved accuracy of medicine reconciliation.**   The perception of the TCAM intervention by participants was that it could reduce mistakes and errors in the dispensing and supply of medicines as patients were transferred between health care settings. Participants saw the system as a way of mitigating against error and ensuring that the new MDS for the patient supplied by the pharmacy contained the correct medication.

*Because inevitably I see it all the time when patients are discharged from hospital, we see so many problems with mistakes, errors, prescribing errors, when patients go from one setting to another, when they go from primary to secondary, secondary to primary, you're constantly facing that battle of mistakes in that transition, and the aim really is to make sure it's communicated reliably. So if the chemist are getting an electronic discharge of like a snapshot of exactly what they were discharged with, that takes out that kind of element of, you know, they've got the information there so they can then hopefully act on that quickly, [. . .] So hopefully the next time the blister pack comes out, it will have the correct medication in, not medication from like preadmission. (Hospital Pharmacist_HP4)*

This community pharmacist talked of a process involving a number of checks to ensure the medicines were accurate and that any changes in the patients medication were dealt with in a timely manner. Further checks were carried out after the patient had received their medicine to ensure that 'everything was fine' and to see if further support was needed in the form of an MUR or other counselling.

*. . .when we get the discharge letter on their discharge, we check with the patient if they're home that day. Because sometimes they send the discharge, but they are still waiting for their medicines and they are not back home [. . .] then we check with the surgery if they've received the discharge letter, if there are any changes. And then we request the medication accordingly. According to the supply that they have had. [. . .] But we just then check with the current medication if there are any changes when we receive it. And then after two weeks, if there are any more changes, again we complete the online request and say, yeah, everything is fine and we have received the request. We have checked with the patient, their MUR or whatever needs doing. (Community pharmacist_HP15)*

The accuracy provided through the service was seen as providing *'seamless care'* for patients across primary and secondary care, who received their correct medications on time.

*I think there's multiple impact really, for the patient it's seamless care, which is the aim of the project, it's about seamless transfer of care between the two sectors. For a patient that's what*

*they need, they want to see the medicines on time, they want to see that actually they've not had to chase things up, they're not worried, they're not anxious about getting their medicines. (Community pharmacist_HP16).*

## Discussion

The World Health Organizations Third Global Patient Safety Challenge (2017) "Medication without harm" [1] specifically recommends that there should be improvements in technology and in the transfer of information from hospital pharmacists to primary care (both community pharmacy and general practices) to improve the medication safety of transitions of care. This study has shown that the successful implementation and adoption of a technological solution to enhance communication about medication across health care transitions from hospital to home was dependent upon the technology and a network of people using that technology. Once the technology was introduced it could only be successfully used and sustained if work practices were adapted and relationships between stakeholders built to create a dynamic network. However, in this TCAM service network activity was predominantly between health care professionals with patients and carers disengaged from the network. The material properties of the TCAM technology provided the infrastructure that facilitated the exchange of enhanced information which in turn impacted upon changes in work practices as the new technology was adapted to in everyday practice. In order for this information to be moved around, a dynamic network of actors was required including hospital pharmacy staff, primary care pharmacy staff and patients and carers. Medication safety activities were reported by staff to be enhanced as a consequence of information transfer through the network with errors, delays and inaccuracies in dispensed medicines for patients in receipt of MDS reduced. Our study therefore builds upon the work of Nazar [27] and Ferguson et al [29] but draws from a wider group of stakeholders and builds in the perspectives of patients and carers. As Ferguson et al [29] conclude there is a need for collective work between hospital and community pharmacists for TCAM services to become fully embedded.

### Medication safety

Both hospital and community pharmacists valued the TCAM intervention as a way of improving medication safety through the timeliness and improved accuracy compared to the previous system and that therefore medicines reconciliation and review was improved. Participants reported that TCAM facilitated the correct medicines being made available for patients without delay which avoiding patients waiting anxiously. This relates to previous research by Ferguson et al [29] who found a similar service was much more efficient. Changes that had been made in hospital could it was reported be more accurately reconciled with the new prescription. Community pharmacists received notifications through the system at admission and discharge and valued receiving both notifications. Community pharmacists particularly felt that the additional information provided by the electronic discharge could allow them to understand more about patients and be more holistic in their approach to care. Wilcock [39] has suggested that the benefits to medicine safety from TCAM projects could be realised through the community pharmacy contractual frameworks to better achieve medicines optimisation after discharge and this is a priority of the new community pharmacy contract [40].

In our study some pharmacists alluded to undertaking additional services such as NMS and MURs, but others said they did not. One particularly reflected on how the service led to further informal conversations with patients when they delivered medicines but others discussed how

patients in receipt of an MDS were unlikely to visit the pharmacy and therefore unlikely to be spoken to by a pharmacist. This has consequences for the service in that while it might possibly lead to fewer errors in dispensed medicines it might not lead to increased support and education for patients and their carers. Latif et al [41] in a qualitative study of the NMS service found that policies surrounding this service were framed around a consideration that all patients were alike. It may be with TCAM interventions that consideration needs to be given for different groups of patients and how the service might reach those who are less likely to attend the community pharmacy in person.

## Socio-technical theory

The TCAM intervention studied here provided for new ways of working that adopted new technologies. As a consequence of such work adaptation new relationships were formed or strengthened and these fed back through the network into further adaptation and ongoing sustainability. Community pharmacies welcomed the addition of 'Pharmalarms'. These were small plug in alert devices that could be located in the dispensary and would alert the dispensing team if a new referral was received thus offsetting the need to log into PharmOutcomes™ or into the pharmacies email system. Such an extension of the technology allowed for and shaped work practices. Hutchby [35] has understood technology as offering "affordances" which allow for the shaping of human activity in the interaction with the technology. In this way the possibilities for action are linked to the technology which constrains or allows for certain actions. Much of the work for the transfer of information prior to the introduction of the TCAM intervention was undertaken via fax. The changes to the electronic transfer changed work practices streamlined the system and increased time available for clinical time in both the hospital setting and in community pharmacy. Petrakaki [42] has previously argued that these affordances are an interconnection between the technology and the organisational systems. Highlighted in this present study are the different technologies available (faxes, the electronic transfer of admission and discharge notes and the MDS themselves) and how they were interconnected to the work that was undertaken and in the adaptation of work practices. Furthermore, the affordances made available by the technology here were seen to strengthen the value of community pharmacy and allow for development of clinical work. Where in some previous literature technology has been seen to reduce professional autonomy and undermine clinical activity [32] this concords with findings of Petrakaki [42] who found that affordances embedded in the material properties of a patient record system could redistribute clinical work and with Jeffries et al [43, 44] who found that practice-based pharmacy clinical activity could be enhanced through a technological solution.

## Fragmentation of the network

Previous work on transfer of care to community pharmacy [29] found that the movement of information in the network was linear from hospital settings to community pharmacy. This concords with what was found in this study to an extent. However, whilst there was fragmentation in the network, the service was perceived as potentially improving communication between general practices and community pharmacy. There were also aspirations for inter-referrals between practices and pharmacies. This could have been strengthened in that Salford has a strong base of practice-based pharmacists and a Neighbourhood Integrated Practice Pharmacist Service (NIPPS) [44] that centrally provides pharmacist support to practices in the area. Participants reflected that involving this group would strengthen communication and the potential of the network.

Patients were also not routinely involved in the network. This could have been because of the nature of the patients who were in receipt of their medicines in an MDS. This group of patients were often older, had a number of long-term conditions and were frequent attenders at hospital. As already stated, this possibly led to fewer interactions between patients and community pharmacists. For some patients, the MDS were delivered and they had little or no connection to the pharmacist. As a consequence, patients and carers felt they did not receive full information about their medicines. The challenges of patient involvement in care, particularly older patients, has been previously reported by Murray et al [45] who explored in a systematic review of international qualitative studies, how patients enact involvement in care in transitions from hospital to home. In keeping with our study they found patients feeling excluded, initiating contact and acting autonomously. Similarly, in a qualitative meta-summary of the experiences of older patients in the discharge process found that older patients had few opportunities to participating in shared decision making [46]. It has been suggested previously that community pharmacy, as opposed to general practice, is often more accessible to patients allowing patients a space to talk openly about medicines but that previously barriers existed in a lack of access to patient medical information for the community pharmacist [47]. As was suggested by pharmacist here the TCAM service could potentially provide further enhanced opportunities for communication with patients as part of a joined -up approach to their care.

## Strengths and limitations

This study strengthens our understanding of how the implementation of technology involves dynamic social processes encompassing a complex network of the human actors utilising the technology. Drawing upon sociotechnical theory with SST has enabled a deeper understanding of these processes

This study is one of very few qualitative studies of TCAM services and the only one exploring the use of the PharmOutcomes™ platform. A strength of this qualitative study was the wide range of stakeholder views including those of patients and their carers. Qualitative research does not aim to be generalisable and this research was conducted as a case study in one geographical area with one main hospital. However, these findings may be framed in context and used to guide other organisations wishing to implement/evaluate similar services. In particular the findings relating to connectivity between different stakeholders may be transferable to other settings and contexts. This may be useful in understanding similar or other interventions that draw across healthcare sectors and involve different professional groups."

The research in this study relied exclusively upon semi-structured interviews. Whilst this was a useful tool to capture views and perspectives, future research using non-participant direct observations might capture how work practices were changed rather than relying upon reported behaviours as previous ethnographic studies have done [48]. An important limitation of our work was that few patients and carers took part in the interviews relative to health professionals. Of the 23 interviews, 5 were with patients and carers. This made exploring the patient perspective more challenging. Several problems were encountered in the recruitment of patients, which included their ability to take part due to frailty and or ill health and due to the fact that many patients who use MDS are older and may be more likely to be physically and emotionally fragile following hospital discharge. For some patients this also meant there were physical barriers such as sight problems that meant reading study information was difficult even when this was provided in large print versions. Some patients also declined because they did not feel their opinion mattered and thus potentially valuable views were lost. Future studies involving such a cohort of patients might usefully consider approaches to empower them to engage in their care and contribute to research.

### Recommendations for further research, policy and practice

As TCAM interventions are rolled out across England it will be important that their implementation is further explored [23]. As the service moves forward it would be of benefit to understand further connectivity between different stakeholders, how information is to be transferred and how networks of relationships can be built to act upon that information to deliver medication safety activities. The intervention here had a linear transfer of information from hospital to community pharmacy but as we have seen this could be enhanced with greater connectivity. Our findings suggested that patients were not fully engaged in the service so it will be of value in the future to look at ways in which patients might become active partners within other networks in similar contexts. The enhancement of clinical practice-based pharmacists through the NHS England initiative may well align with further connectivity between secondary and primary care that our study found as being important for medicines reconciliation and optimisation. [49, 50] This could involve further integration of shared records between hospital, community and practice-based pharmacists and further two-way communication between primary and secondary care. The TCAM service evaluated here was specifically designed for patients who received their medicines in a pre-prepared MDS but the service could include other patients with multi-morbidity who were not in receipt of an MDS or patients from specific groups such as those who had had a cardiac event or were in substance misuse programmes and were in receipt of prescriptions for methadone. As services are extended to include such other patients it will be helpful to see how the service works at local levels through further qualitative evaluation and if the findings her transfer to those contexts.

In addition, further research is needed to explore the extent of pharmacist activity in response to the electronic referrals. Mixed methods approaches have been utilised to examine the implementation of information technology for medication safety [43] and could be utilised here to unpick the wider picture as well as the detail of the intervention and frame the context of the use of the system.

## Conclusions

The TCAM intervention was perceived as effective in providing community pharmacists with timely, more accurate and enhanced information upon discharge of patients in receipt of an MDS. This supported pharmacy teams to establish and/or strengthen networks in order to provide clinical services designed to ensure that accurate medication reconciliation was completed, and the correct medication was dispensed for the patient. This evaluation also indicated potential improvements in medication safety and streamlined work practices through increased clinical and decreased administrative time. However, the intervention was implemented into a complex and at times fragmented network, and we recommend opportunities be explored to fully integrate this network to involve patients/carers, general practice pharmacists and two-way communication between primary and secondary care to further enhance the reach and impact of the TCAM service.

## Supporting information

**S1 Checklist.**
(DOCX)

**S1 Appendix. Interview topic guide.**
(DOCX)

**S2 Appendix. First coding framework.**
(DOCX)

**S3 Appendix. Final coding framework.**
(DOCX)

**S4 Appendix. Data set -all extracts from final coding.**
(PDF)

## Acknowledgments

We are grateful for the help from Lindsay Harper, Director of Pharmacy at Salford Royal NHS Foundation Trust. We are grateful to all interview participants who kindly gave their time.

## Author Contributions

**Conceptualization:** Mark Jeffries, Richard N. Keers, Hilary Belither, Caroline Sanders, Kay Gallacher, Fatema Alqenae, Darren M. Ashcroft.

**Data curation:** Mark Jeffries, Richard N. Keers.

**Formal analysis:** Mark Jeffries, Richard N. Keers, Hilary Belither, Kay Gallacher, Fatema Alqenae, Darren M. Ashcroft.

**Investigation:** Mark Jeffries.

**Methodology:** Mark Jeffries, Richard N. Keers, Hilary Belither, Caroline Sanders, Darren M. Ashcroft.

**Project administration:** Mark Jeffries, Richard N. Keers, Hilary Belither.

**Supervision:** Darren M. Ashcroft.

**Validation:** Mark Jeffries, Richard N. Keers, Hilary Belither, Caroline Sanders, Kay Gallacher, Fatema Alqenae, Darren M. Ashcroft.

**Writing – original draft:** Mark Jeffries.

**Writing – review & editing:** Mark Jeffries, Richard N. Keers, Hilary Belither, Caroline Sanders, Kay Gallacher, Fatema Alqenae, Darren M. Ashcroft.

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
