## [Decision Letter · Decision Letter 0]

17 Feb 2021

PONE-D-20-30293

Understanding the implementation, impact and sustainable use of an electronic pharmacy referral service at hospital discharge: A qualitative evaluation using Strong Structuration Theory

PLOS ONE

Dear Dr. Jeffries,

Thank you for submitting your manuscript to PLOS ONE. After careful consideration, we feel that it has merit but does not fully meet PLOS ONE’s publication criteria as it currently stands. Therefore, we invite you to submit a revised version of the manuscript that addresses the points raised during the review process.

Please note that one reviewer recommended Rejection. After considering their comments we feel it may be possible to address these concerns with a thorough revision. However, please bear in mind that further progress will depend on thoughtful responses to these comments.

We look forward to receiving your revised manuscript.

Kind regards,

Bradford Dubik

Academic Editor

PLOS ONE

Journal Requirements:

Additional Editor Comments (if provided):

Reviewers' comments:

Reviewer's Responses to Questions

**Comments to the Author**

1. Is the manuscript technically sound, and do the data support the conclusions?

Reviewer #1: Partly

Reviewer #2: Yes

2. Has the statistical analysis been performed appropriately and rigorously? 

Reviewer #1: N/A

Reviewer #2: N/A

3. Have the authors made all data underlying the findings in their manuscript fully available?

Reviewer #1: Yes

Reviewer #2: No

4. Is the manuscript presented in an intelligible fashion and written in standard English?

Reviewer #1: No

Reviewer #2: Yes

5. Review Comments to the Author

Reviewer #1: Very interesting study and useful empirical findings.

As one of the people who developed technology-enhanced strong structuration theory (TESST), I'm intrigued by how it was applied in this case. I can see the authors have broadly understood the principles behind it and have referenced relevant work. However I'm a bit worried that it hasn't been applied in quite the way we intended. The key unit of analysis in SST (and also in TESST, the technology-enhanced version) is the *conjuncture* - that is, a small-scale social situation such as an encounter between a patient and a pharmacist.This would require ethnography. What's been done instead is a set of interviews in which situations are talked about in the abstract (pharmacists, for example, talk about patients in general rather than a particular incident with a particular patient). The really strong SST or TESST studies I've seen have all done micro-analysis of interactional talk and action in *real* social situations. I accept that this may not have been possible (or thought of in advance), but it does limit the study significantly. In addition, for a rigorous TESST analysis, we really need a *material* analysis of the technologies to ask what social structures have been inscribed in them - for example, how has the software 'configured' the patient (what assumptions about patienthood have been built into the algorithms and how do these features and functions of the technology shape and constrain what happens?).

To be honest I think the empirical findings are pretty interesting without SST or TESST - so I think there are two options here: leave out the attempt to use this particular theory and redraft the paper as a simple thematic analysis, OR go for the more theoretically-rich redraft with a bit more analysis (eg. go through the data and find when people are talking about concrete cases and focus the analysis on those). Depends whether your main goal is to contribute to theory or improve the NHS!

One further suggestion - the paper would benefit greatly from a general edit to sharpen the message. Abstract for example could be condensed quite a bit.

Hope this helps improve the paper.

Trish Greenhalgh

Reviewer #2: Summary

This is an interesting text, in an important and pressing area of research: the assessment of technology solutions applied to healthcare. In this case an electronic pharmacy data management system, with potential to support community pharmacies to track/trace patients medication history when admitted/discharged into/from hospital. An evaluation of the TCAM system was undertaken for patients who receive their prescribed medication via Monitored Dosages Systems to understand it adoption and functionality from the end user perspective, healthcare professional and patient/carer alike.

With an increasing pressure on healthcare systems, the use of information technology data management systems is a vital area to ensure clear and reliable communication across and within organisations that manage patient care. An ageing population gives rise to an increase in the number of patients suffering from physical and mental frailty, and systems such as TCAM for MDS can allow data to be joined up and lead to more efficient and effective care for this group.

Major Issues

There do not appear to be any major issues with this article, which reports some interesting work, presenting and assessing the value of pharmacy electronic data management systems, that operate across organisational boundaries. The study design, research questions and data collected to provide an answer are appropriate.

However The following areas may be worth further reflection:

Background

1. It could be interesting to refer to other E-prescriptions solutions, and why TCAM was chosen

2. It is worth saying more about how TCAM works, and reporting on it in more detail.

3. Is it worth outlining other frameworks considered (other than ANT), and saying why they were discarded and SST selected as best one to use?

Data Analysis

1. Could the data be described in more detail. How many hours of interviews were there in total? What were the job titles or those interviews. Could this be presented as a table

2. It could be helpful to include the interview topic guide/guides, maybe as an appendix. Page10 Lines 192-198 refer to an interview guide, but only gives broad areas interviews were interested in, which all seem to be focused on professionals, not patients/carers.

3. Can more be said about the participants that declined to take part (page 34, line 748). What about professionals who declined.

4. Could the analysis be clearer and reported to show how the themes were present across the data set. Overall is could be good to understand more about the coding process and derivation of themes in general.

5. It may also be supportive of the research findings to include the coding framework that was developed. The authors talk of thematic analysis, and refer to 4 main thematic themes discussed in the results section. Where there any other themes of sub-themes of interest, it could benefit the design of further research to have access to these.

6. The authors give 4 main themes (and sub-themes?) as the foundation of the results of the data analysis, broadly speaking these are:

○ The Network, implementation, use and sustainability

§ Relationships between different health professionals and what this achieves

§ Communication and movement of information in the network

§ Patients and carers (this seems more to fit in with communication- or lack of it, from the patients perspective)

○ External factors and (Technology/materiality)

§ The previous system and changes in technology

§ New technology provided new infrastructure

○ Internal factors, changes to work practice (organisational/social)

§ Adaptation of work processes to the availability of information

§ Timesaving, speed, and efficiency

○ Outcomes and Improvements (Agency)

§ Clinicians have more time

§ There are less prescribing errors

Each of main theme is presented as a section, with sub-headings (presumably corresponding to sub-themes). but it is at times difficult to develop a clear picture of how these were arrived at and for the reader to chart a course through the narrative in this format. Could themes be presented/summarised as a table or diagram to give greater clarity.

7. Were these the only themes that emerged, how did they compare between the different interview subject types, a lot of what is reported seems from the perspective of the professional, perhaps this is understandable given they provided more data. Could community pharmacists provide indirect feedback from patients regarding the system?

8. What techniques for "trustworthiness" (reliability) in the analysis were used (inter-rata coding, etc)?

9. What are the boundaries of the research findings ("transferability"). Page 34 Line 734 says the study did not aim to be generalisable, but some of the findings may well be, especially if echoed from more than one participant (triangulation).

Minor issues

1. Title. Could '…and sustainable value…' be a clearer word than 'sustainable use', which seems assumed in the concept of sustainability? Given only the TCAM is evaluated, could this not be in the title? (i.e. "The TCAM electronic pharmacy referral service at hospital discharge: A qualitative evaluation using Strong Structuration Theory")

2. The page numbers on the COREQ checklist do not seem to match the article (25 and 26 refer to the coding and derivation of themes as being on pages 11 and 10 respectively, however these could not be found there.

3. Page 11 Line 209. The payment for participation seems quite considerable, was this raised at ethics? Why were staff paid more than patients? I am not sure that specifying the actual amounts benefits the article.

4. While clearly data should remain unidentifiable, but could participant demographic data (gender, age, etc) yield some insights, especially in terms of subsequent meta-analysis and research into technology adoption.

5. It appears TCAM can apply to admission and discharge. In the title is says this is about discharge, but some places this is not always clear (Page 22 465-473 , the pharmacists seems to be taking about admission). Can this duality worth further exposure in the discussion and conclusions? Could understanding how they are different be helpful?

Conclusion

Overall this work should be given full consideration for publication, especially given there is so little published in terms of TCAM, or the evaluation of pharmacy electronic data systems, from the perspective of patient drug safety at discharge. Socio-technical Systems and Strong Structuration Theory, are ideal to provide an appropriate lens to better understand the interface of healthcare information technology and the cultural, social and organisational aspects of shared work activity. Their reporting will also allow to build up the evidence base of socio-technical theory based research in healthcare and provide a focused perspective on this challenging area.

6. PLOS authors have the option to publish the peer review history of their article (what does this mean?). If published, this will include your full peer review and any attached files.

Reviewer #1: **Yes: **Trish Greenhalgh

Reviewer #2: No

---

## [Author Response · Author response to Decision Letter 0]

23 Apr 2021

PONE-D-20-30293

Understanding the implementation, impact and sustainable use of an electronic pharmacy referral service at hospital discharge: A qualitative evaluation using Strong Structuration Theory

Dear Dr Dubik

Re: ‘Understanding the implementation, impact and sustainable use of an electronic pharmacy referral service at hospital discharge: A qualitative evaluation using Strong Structuration Theory

Thank you for the thoughtful and constructive feedback on our manuscript, which we have carefully considered. We have in the light of this feedback made a number of revisions to the manuscript, which are detailed in the table below.

In response to Reviewer One’s very helpful comments we have decided to align the paper not to Strong Structuration Theory but to broader Sociotechnical Theory. We feel that this provides a clearer focus and interpretation of our findings from the thematic analysis that we undertook. As a consequence, we have, in addition to changes to the manuscript, made changes to the title. This now reads: Understanding the implementation, impact and sustainable use of an electronic pharmacy referral service at hospital discharge: A qualitative evaluation from a sociotechnical perspective.

We are confident that these revisions will be to your satisfaction, but please do contact me if you require any further clarification or revisions. 

Yours sincerely,

Mark Jeffries, on behalf of the co-authors

 

Editor comments Response/Changes made Location of change in tracked changed manuscript

Reviewer One 

As one of the people who developed technology-enhanced strong structuration theory (TESST), I'm intrigued by how it was applied in this case. I can see the authors have broadly understood the principles behind it and have referenced relevant work. However I'm a bit worried that it hasn't been applied in quite the way we intended. The key unit of analysis in SST (and also in TESST, the technology-enhanced version) is the *conjuncture* - that is, a small-scale social situation such as an encounter between a patient and a pharmacist. This would require ethnography. What's been done instead is a set of interviews in which situations are talked about in the abstract (pharmacists, for example, talk about patients in general rather than a particular incident with a particular patient). The really strong SST or TESST studies I've seen have all done micro-analysis of interactional talk and action in *real* social situations. I accept that this may not have been possible (or thought of in advance), but it does limit the study significantly. In addition, for a rigorous TESST analysis, we really need a *material* analysis of the technologies to ask what social structures have been inscribed in them - for example, how has the software 'configured' the patient (what assumptions about patienthood have been built into the algorithms and how do these features and functions of the technology shape and constrain what happens?).

To be honest I think the empirical findings are pretty interesting without SST or TESST - so I think there are two options here: leave out the attempt to use this particular theory and redraft the paper as a simple thematic analysis, OR go for the more theoretically-rich redraft with a bit more analysis (eg. go through the data and find when people are talking about concrete cases and focus the analysis on those). Depends whether your main goal is to contribute to theory or improve the NHS!

 Thank you for these helpful comments. We have decided, as you suggest, to leave out the connection to Strong Structuration Theory and to focus the paper as a thematic analysis broadly informed by sociotechnical theory. We feel that the results as reported remain strong without the additional lens of SST. We also feel that the section on sociotechnical theory that reflects how the processes of implementing the technology shaped work practices remains valid in our redrafted version. We have made a number of changes in the manuscript as detailed in the next column and in the Revised Manuscript with Track Changes. These changes include the title which now reads: 

Understanding the implementation, impact and sustainable use of an electronic pharmacy referral service at hospital discharge: A qualitative evaluation from a sociotechnical perspective. Abstract – Methods section page 2

Abstract – Results page 2-3

Background pages 6-9.

Methods pages 11-12.

Methods data analysis pages 12.

Results page 12-13, p24.

One further suggestion - the paper would benefit greatly from a general edit to sharpen the message. Abstract for example could be condensed quite a bit. We have revised the abstract and made changes throughout the manuscript to provide greater clarity and be more concise. In particular we have truncated some of the quotations in the results. 

Reviewer Two 

Background 

1. It could be interesting to refer to other E-prescriptions solutions, and why TCAM was chosen

 Thank you for this comment. Transfers of Care Around Medicines (TCAM) was chosen because of its importance both nationally across the NHS in England and locally to the hospital where we conducted the research. TCAM is now part of the NHS Discharge Medicines Service introduced in January 2021. TCAM is being rolled out nationally across England. TCAM is not solely e-prescribing service but a method of transferring electronic messages about patients on discharge from hospital to their usual community pharmacist, which may include a discharge prescription. TCAM operates utilising the electronic prescribing and medicines administration systems in hospitals. The particular service we evaluated in addition utilised the PharmOutcomes platform a commonly used in community pharmacies. We have added some more detail about TCAM and included two new references to policy documents in the introduction. We have removed the information about the intervention from the box on page 8 and rewritten this placing it in the Methods. Pages 5-6. Page 10

2. It is worth saying more about how TCAM works, and reporting on it in more detail.

 We have added detail to the policy landscape in which TCAM is situated as detailed in our response to your point 1 above. We have made specific changes to the details about the intervention, now placed in the methods. We have removed detail about the ways in which different health professionals were involved in the intervention and the basic outline of it and placed these in the introduction on page 8. We have given a clearer description of the specific processes involved in the intervention in the methods. Introduction Pages 5,6.

Page 8.

Methods

Page 10

3. Is it worth outlining other frameworks considered (other than ANT), and saying why they were discarded and SST selected as best one to use? We have removed references to strong structuration theory and are no longer aligning the paper to this theory but to broader and wider sociotechnical theory. Please see our response to Reviewer 1’s points above. n/a

Data Analysis 

1.Could the data be described in more detail. How many hours of interviews were there in total? What were the job titles or those interviews. Could this be presented as a table We have added a table (Table 1) detailing the interviews and participants. Cited on page 12. table at end of manuscript

2. It could be helpful to include the interview topic guide/guides, maybe as an appendix. Page10 Lines 192-198 refer to an interview guide, but only gives broad areas interviews were interested in, which all seem to be focused on professionals, not patients/carers. We agree that this would be useful. We have included the topic guide as a supplementary file. Supplementary files

3. Can more be said about the participants that declined to take part (page 34, line 748). What about professionals who declined. We have added these two sentences to details to the ‘Sampling and Recruitment’ section of the methods. ‘Three health professionals declined to take part or did not return consent to contact forms. Those who declined cited workload and time commitments as the reason for their non-participation.’ We have left the detail regarding the patients who declined to take part in the section on ‘Strengths and Limitations’ because we feel it is best placed here to highlight how similar difficulties in recruitment from similar cohorts of patients might need to be addressed in future studies. Methods -Sampling and Recruitment page 11

4. Could the analysis be clearer and reported to show how the themes were present across the data set. Overall is could be good to understand more about the coding process and derivation of themes in general. We have rewritten the data analysis sub-section of the methods – This now reads: 

“Following transcription, anonymised interviews were organised using QSR NVIVO® Pro v12 software. We undertook a thematic analysis informed by Braun and Clark [35]. Analysis followed an iterative approach and was concurrent with data collection. This allowed for the development of a coding framework and for emergent findings to be explored in subsequent data collection. MJ read each transcript in a process of immersion. A selection of early interviews was read and discussed by MJ, RNK, KG, HB and FA. MJ inductively coded a sample of six transcripts focusing upon the interactions of different people in the network, the social processes within the intervention and the changes brought about by the technology. Identifying these features, and patterns allowed for groups of codes and potential themes to be refined into a coding template with codes grouped into sets (see supplementary file S1). This coding template, and further coded transcripts, where then discussed by MJ, RNK, KG, HB and FA. From these discussions the template was revised and refined into themes and codes (see supplementary file S2). This template was then applied to the full dataset. It was from this final stage of the data analysis that the final themes and sub-themes were interpreted (see table 2).”

 Methods – Data analysis pages 14-15

5. It may also be supportive of the research findings to include the coding framework that was developed. The authors talk of thematic analysis, and refer to 4 main thematic themes discussed in the results section. Where there any other themes of sub-themes of interest, it could benefit the design of further research to have access to these. Thank you for this point. We agree that this may be useful and have added the coding frameworks as a supplementary files. Supplementary files

The authors give 4 main themes (and sub-themes?) as the foundation of the results of the data analysis, broadly speaking these are:

○ The Network, implementation, use and sustainability

§ Relationships between different health professionals and what this achieves

§ Communication and movement of information in the network

§ Patients and carers (this seems more to fit in with communication- or lack of it, from the patients perspective)

○ External factors and (Technology/materiality)

§ The previous system and changes in technology

§ New technology provided new infrastructure

○ Internal factors, changes to work practice (organisational/social)

§ Adaptation of work processes to the availability of information

§ Timesaving, speed, and efficiency

○ Outcomes and Improvements (Agency)

§ Clinicians have more time

§ There are less prescribing errors

Each of main theme is presented as a section, with sub-headings (presumably corresponding to sub-themes). but it is at times difficult to develop a clear picture of how these were arrived at and for the reader to chart a course through the narrative in this format. Could themes be presented/summarised as a table or diagram to give greater clarity. We agree that this would be helpful. We have added a table (table 2). Please also see our response to your point 4 above.

 Page 15; table 2 at end of manuscript

7. Were these the only themes that emerged, how did they compare between the different interview subject types, a lot of what is reported seems from the perspective of the professional, perhaps this is understandable given they provided more data. Could community pharmacists provide indirect feedback from patients regarding the system? We hope that the table of themes and supplementary files we have provided show how the development of themes occurred. We accept that there was a stronger perspective from health professionals in the results. We have reflected upon this in the limitations of the study at page 35 and respectfully draw your attention to that section. Whilst the patient perspective is we agree, less prominent, there is a long section in the results that covers the perspective of patients and carers. Pharmacists did not directly comment about patients views of the intervention but on page 22 at the end of the section on patients and carers, we do give a quotation from one pharmacist who talked of patients valuing the additional communication they received from the pharmacist as a consequence of the TCAM intervention. We added a sentence before this quotation which highlights this further. Table 2, supplementary files. Page 22

8. What techniques for "trustworthiness" (reliability) in the analysis were used (inter-rata coding, etc)? The transcripts, coding, coding frameworks and interpretated themes were discussed at length by co-authors as detailed in the newly worded data analysis section and detailed above. Page 14

9. What are the boundaries of the research findings ("transferability"). Page 34 Line 734 says the study did not aim to be generalisable, but some of the findings may well be, especially if echoed from more than one participant (triangulation). We agree that this needs more clarity. We have added the following sentence to the discussion “In particular the findings relating to connectivity between different stakeholders may be transferable to other settings and contexts. This may be useful in understanding similar or other interventions that draw across healthcare sectors and involve different professional groups.” 

We do reflect upon our findings in the section on implications for further research, policy and practice. We have made some minor changes here to further emphasize the potential transferability of the findings. Discussion Page 35, Page 36

Minor Issues 

1. Title. Could '…and sustainable value…' be a clearer word than 'sustainable use', which seems assumed in the concept of sustainability? Given only the TCAM is evaluated, could this not be in the title? (i.e. "The TCAM electronic pharmacy referral service at hospital discharge: A qualitative evaluation using Strong Structuration Theory") We have changed the title in response to reviewer one’s comments. The transfer of Care around Medicines service is a broad service across England. The e-referral service we evaluated was one TCAM intervention but was specific to the local hospital trust. We trust our changes to the description of the intervention in the methods section help make this clear. We therefore feel the title should not mention TCAM. Title

2. The page numbers on the COREQ checklist do not seem to match the article (25 and 26 refer to the coding and derivation of themes as being on pages 11 and 10 respectively, however these could not be found there. We have made changes to the COREQ so that the page references are accurate. Thank you. Please see attached COREQ

3. Page 11 Line 209. The payment for participation seems quite considerable, was this raised at ethics? Why were staff paid more than patients? I am not sure that specifying the actual amounts benefits the article. In our experience it is useful to specify payment amounts to participants for the purposes of transparency. Staff were paid £50 to incentivise participation and to compensate for the time they were required to be away from work. In the case of community pharmacists particularly they had to leave the dispensary to undertake the interview. Payments were in line with INVOLVE rates. 

4. While clearly data should remain unidentifiable, but could participant demographic data (gender, age, etc) yield some insights, especially in terms of subsequent meta-analysis and research into technology adoption. We do not feel that gender played a part in the responses given and wish the gender neutrality of participants to be preserved. Participants were not asked their age or gender so we cannot report this. We feel that job titles were more important and these are reported in the quotations and in table 1. Results and table 1

5. It appears TCAM can apply to admission and discharge. In the title is says this is about discharge, but some places this is not always clear (Page 22 465-473 , the pharmacists seems to be taking about admission). Can this duality worth further exposure in the discussion and conclusions? Could understanding how they are different be helpful? We agree that the service is framed as being around discharge but in fact community pharmacists received a notification of the patient being admitted to hospital. We hope that changes we have made in response to comments above help clarify this. These changes are: the change to the title of the manuscript and the changes to the description of the intervention in the methods. In addition we have added to the discussion the following sentence: “Community pharmacists received notifications through the system at admission and discharge and valued receiving both notifications.”

 Methods – The InterventionPage. 10 Discussion Page 32

---

## [Decision Letter · Decision Letter 1]

29 Nov 2021

Understanding the implementation, impact and sustainable use of an electronic pharmacy referral service at hospital discharge : A qualitative evaluation from a sociotechnical perspective.

PONE-D-20-30293R1

Dear Dr. Jeffries,

We’re pleased to inform you that your manuscript has been judged scientifically suitable for publication and will be formally accepted for publication once it meets all outstanding technical requirements.

Kind regards,

Kathleen Finlayson

Academic Editor

PLOS ONE

Additional Editor Comments (optional):

Please consider the reviewer's suggestion to be a little more succinct in your wording,  e.g., some of the illustrative quotes could be tightened to one or two sentences /parts of sentences

Reviewers' comments:

Reviewer's Responses to Questions

**Comments to the Author**

1. If the authors have adequately addressed your comments raised in a previous round of review and you feel that this manuscript is now acceptable for publication, you may indicate that here to bypass the “Comments to the Author” section, enter your conflict of interest statement in the “Confidential to Editor” section, and submit your "Accept" recommendation.

Reviewer #1: All comments have been addressed

2. Is the manuscript technically sound, and do the data support the conclusions?

Reviewer #1: Yes

3. Has the statistical analysis been performed appropriately and rigorously? 

Reviewer #1: N/A

4. Have the authors made all data underlying the findings in their manuscript fully available?

Reviewer #1: Yes

5. Is the manuscript presented in an intelligible fashion and written in standard English?

Reviewer #1: Yes

6. Review Comments to the Author

Reviewer #1: I'm now happy with it theoretically and empirically but it's a VERY LONG READ! It's an editorial decision but if I were the editor (which I'm not), I'd suggest losing 1000 words to tighten up.

7. PLOS authors have the option to publish the peer review history of their article (what does this mean?). If published, this will include your full peer review and any attached files.

Reviewer #1: **Yes: **Trisha Greenhalgh

---

## [Editor Report · Acceptance letter]

9 Dec 2021

PONE-D-20-30293R1 

Understanding the implementation, impact and sustainable use of an electronic pharmacy referral service at hospital discharge: A qualitative evaluation from a sociotechnical perspective. 

Dear Dr. Jeffries:

I'm pleased to inform you that your manuscript has been deemed suitable for publication in PLOS ONE. Congratulations! Your manuscript is now with our production department. 

Kind regards, 

on behalf of

Dr. Kathleen Finlayson 

Academic Editor

PLOS ONE